# Quantitative Experimental Embryology: A Modern Classical Approach

**DOI:** 10.3390/jdb10040044

**Published:** 2022-10-18

**Authors:** Lara Busby, Dillan Saunders, Guillermo Serrano Nájera, Benjamin Steventon

**Affiliations:** Department of Genetics, University of Cambridge, Cambridge CB2 3EH, UK

**Keywords:** mechanics, induction, regulation, scaling

## Abstract

Experimental Embryology is often referred to as a classical approach of developmental biology that has been to some extent replaced by the introduction of molecular biology and genetic techniques to the field. Inspired by the combination of this approach with advanced techniques to uncover core principles of neural crest development by the laboratory of Roberto Mayor, we review key quantitative examples of experimental embryology from recent work in a broad range of developmental biology questions. We propose that quantitative experimental embryology offers essential ways to explore the reaction of cells and tissues to targeted cell addition, removal, and confinement. In doing so, it is an essential methodology to uncover principles of development that remain elusive such as pattern regulation, scaling, and self-organisation.

## 1. Introduction

The regulation of developmental processes is an inherently multi-scalar problem. One could consider how the action of genes and their regulatory networks operate over time to drive changes in cell state. Such cell state changes impact the dynamic behaviour of cells as they move, change shape, size, or interact with neighbouring cells and the extracellular matrix. Such multi-cellular interactions impact the physical properties of tissues, their rate of expansion and/or their shape. Tissues then interact with one another through a combination of mechanical or chemical signalling interactions to build embryos and organisms. At the same time, each of these levels of biological organisation receives multi-scale feedback interactions, as tissue morphogenesis responds to inductive interactions and multi-tissue mechanics; or as gene regulatory networks respond to cell-to-cell interactions and changes in their mechanical environment. Experimental embryology is an approach of the developmental biology that can be used to probe such multi-scale interactions; particularly in terms of how alterations at higher levels of organisation impact those at lower levels via downward causation. Experimental embryology is considered a classical approach due to its long history in the field that pre-dates the introduction of molecular biology techniques to the developmental biologist’s toolkit [1]. However, we would argue that its utility in investigating highly timely questions in the field such as scaling, self-organisation, developmental timing, and the mechanics of morphogenesis, sets it at the very forefront of developmental biology. Moreover, the increasing use of quantitative read-outs of such manipulations renders quantitative experimental embryology into a modern classic.

Broadly speaking, experimental embryology can be categorised into three types of manipulation, which we address in this review’s three parts (Table 1). Firstly, the addition of cells to a new region of the embryo, or as part of a conjugate of tissues in vitro. Secondly, the removal of cells through targeted cell or tissue ablations to probe the mechanisms of pattern regulation and regeneration. Finally, through altering the mechanical environment of cells through in vitro cell culture and tissue confinement. Each of these approaches have been applied to great effect in the study of neural crest development, both in terms of its early specification and its later migration. Through making use of the specific advantages of the *Xenopus laevis* embryo, the laboratory of Roberto Mayor has been able to use experimental embryology approaches to determine the rules by which cells, cellular collectives, and tissue interact with one another during neural crest development. Importantly, these finding are always linked back to in vivo experiments to determine whether the newly uncovered mechanisms of neural crest development hold true for cells in their normal environment. Taking this approach as further inspiration to motivate a re-focus to experimental embryology as a core tool of the developmental biologist, we aim to review a range of contemporary examples of its application (Table 2). Our aim is not to review all such examples, as this would take us beyond the scope of a single review, but instead to provide the interested reader with a selection of quantitative experimental embryology applications that communicate the breadth of its application.

## 2. Cell Addition as an Essential Tool in Experimental Embryology

Experimental embryologists have added cells to embryos for many years in the form of transplantation (grafting) experiments, with these types of manipulations yielding influential results including the conception of organizer regions within the embryo [19], as well as providing methods for fate mapping [20]. In this section of the review, we will consider two key examples of experiments which have informed our understanding of regulative development: firstly, the role of inter-embryo cell combinations (chimaera formation) in demonstrating the remarkable regulative ability of mammalian embryos, and secondly the specific example of inductive reprogramming. The key difference between these two examples is that in the first instance, the cells being added are often equivalent (in stage and potency) whilst in the other the cells being added are distinctly different, being capable of inducing a response in the host tissue. We will argue that the two types of experimental embryology manipulations described parallel natural instances of cells being added to the early embryo and discuss outstanding questions in this field. Modern molecular approaches show great promise in delving into the underlying mechanisms which allow regulation of development when additional cells are added, and given the spontaneous natural occurrence of these events, may have substantial implications for medicine and fertility. 

### 2.1. Chimaeras, Homotypic Grafts and Size Regulation

In Greek mythology, a chimaera was a creature within which three distinct creatures were spliced together: the head of a lion, the body of a goat, and the tail of a dragon [21]. In our modern-day lexicon, the word has come to be more medically relevant, being used to refer to any living creature which contains at least two genetically distinct cells. Chimaeras occur naturally in both humans and other animals, though many go completely undetected within the animal’s lifetime due to a lack of obvious effects (adverse or otherwise). In a small subset of human cases, the medical literature has described phenotypic effects of chimerism that can result, including the first documented instance of human chimerism in which a patient presented with a mixed blood-type [22]. Other described effects include non-matching eye colours, intersex phenotypes, and patchy skin: all of which can result from the combination of two genetically distinct types of cells within the same individual [23]. These phenotypes can occasionally have negative impacts including infertility or psychological effects.

Depending upon the relative number of ‘foreign’ cells in an individual, naturally occurring chimaeras can be classified as micro chimaeras or tetragametic chimaeras, which have distinct natural origins during gestation. A micro chimaera is an individual in which a small number of cells within the body have a distinct genetic make-up and are produced during gestation through the transfer of a small number of cells either between mother and foetus (in either direction), or in some cases between siblings [23]. In contrast, a tetragametic chimaera contains a substantial number of cells of both genotypes and may form through two distinct fertilisation events and subsequent in utero fusion of the zygotes/early embryos [23]. This event has been previously described as involving “a disappearing twin”. The details of natural chimaera formation go beyond the scope of this review but are mentioned here due to the striking parallels they have to experimental embryological manipulations. We should note the importance of viviparity in facilitating these exchanges of cells between individuals during gestation; in reproductive modes where the embryo develops outside of the parent within a protective covering, opportunity for these events will be scarce if not absent.

Though these examples from medicine are not outputs of experimental embryology, they do closely mirror the kinds of situations in which experimental embryologists have added cells to embryos (Figure 1). Figure 1 conceptualises the examples that we will discuss of adding cells to embryos along two axes: degree of naturality, and the number of foreign cells being added. The examples from human medicine discussed fall on the ‘natural’ side of the scale, but we will now turn to consider the parallel examples in experimental embryology.

### 2.2. Blastula Aggregation

In 1961–1962, two independent researchers combined two murine embryos at the 8-cell stage, finding that the resulting aggregate (chimaera) was able to undergo normal development and lead to viable offspring [24,25]. Subsequently, other groups have extended this result to much larger embryonic aggregates, for example, giant blastocysts can be produced through the combination of 5–9 embryos and develop normally [26]. The success of these aggregation experiments, now exploited widely in mouse research to produce transgenic or mutant offspring, were—and remain—remarkable. For the chimeric embryo to develop to term and produce appropriately sized offspring requires a mechanism of size regulation: the ability to sense, and respond to, a perturbation in the size of the embryo. Note that an additional feature must be present—the appropriate allocation of cell fate, but this will not be discussed in detail here. The response of the embryo to size perturbation was shown by Lewis and Rossant (1982) [27] to be achieved via the modulation of cell cycle length. Neither apoptosis nor the dividing populations of cells appeared altered in the aggregates, but the dividing cells in these embryos had altered dynamics relative to control embryos [27]. Importantly, control embryos experienced a burst of proliferation at 6 days and 8 h post-coitus that was absent from the double-sized embryos [27]. Thus, the murine embryo size regulates upon addition of cells by modulating the proliferative dynamics of its cells. 

These results go some way to explaining the downstream mechanism of size regulation once a perturbation is sensed, but do not explain how a cell senses that it is in a larger or a smaller embryo. More recently, possible molecular inputs that could account for this sensing step have been described. Application of saturating concentrations of Platelet-Derived Growth Factor (PDGF), a mitogen, to embryonic stem cells in culture can influence cell division. Whilst in vivo PDGF levels will gradually decline as cell populations increase in number, application of high PDGF in culture, or in transgenic embryos, is able to artificially stimulate cell division regardless of the prior division state of cells in the embryo [28]. However, these experiments were conducted with spinal cord progenitors at a much later stage of development than those considered in the aggregation experiments above. It remains to be seen what this cue might be in the context of the morula stage embryo. 

Whilst experimental embryology reveals the impressive regulative capacity of mammalian embryos, 35% of embryonic cells are removed during normal development of the mouse [29]. Thus, the observation of regulative and size-robust mammalian development is possibly unsurprising in the context of natural loss of cells and hence plastic cell numbers in early development. Work on cell competition has shown the importance of molecular pathways involving mTOR and cMyc in these processes. When cells that normally show a loser phenotype (e.g., 4N cells) are manipulated to have higher mTOR pathway activity, they are rescued from their usual apoptotic activity and retained in the population [29]. When these cells are injected into the early mouse embryo to create a chimera, they show enhanced competitive abilities and are not eliminated from the population. The mTOR pathway, responsible for aspects of cell competition, was shown to be under the control of p53 in this context, and the removal of p53 activity is able to create ‘super-competitor’ cells [29,30]. Similarly, cMyc protein levels correlate with competitiveness, and over-expression is able to direct the elimination of otherwise healthy cells from the population [31]. Together, these studies have provided insight into how less fit cells can be eliminated during early development, with the implication that when cells are eliminated, the embryo must be able to size-regulate. As mentioned, what is still not clear in this system is how the remaining cells in the population sense population size. This is vital for effective size regulation, and we would argue that this is an outstanding important question in the field. Experimental embryology in combination with modern molecular methods is a promising approach to address this question.

A recent example that illustrates this promise comes from single-cell transcriptomic analysis of Brachyury (T) mutant chimeric embryos [3]. The combination of cells with distinct genotypes in the early mouse embryo to form a chimera is a powerful approach to understanding the proclivities and behaviours of these cells; in particular, this system allows researchers to ask questions about the relative importance of cell-autonomous and non-autonomous effects of these mutations. With the rise of single-cell RNA sequencing technologies in the past few years, it is now possible to assay the whole transcriptome in individual cells, allowing consideration of both differences between wildtype and mutant cell states, but also the heterogeneity that exists within these populations. Guibentif et al. [3] combined experimental embryology—production of *T* mutant chimeric embryos—with single-cell RNA sequencing to show that two distinct populations of somitic progenitor cells exist in the mouse embryo, characterised by high and low T states. This led them to propose that development of the most anterior somites of the mouse embryo occurs independently of T function.

To summarise, the first mouse chimeras, published in the early 1960’s, have had far-reaching effects. The production of a chimera is an example of experimental embryology, representing the addition of cells to the early embryo. This can be on a small scale—the injection of a few cells into the blastocoel—or a large scale—the aggregation of up to 9 early embryos. The success of chimera production in the mouse, as well as other vertebrate species, is not an unremarkable feat, but can be rationalised in the context of more recent data, including the substantive natural exclusion of cells with fitness defects from the early embryo [32]. Chimera formation has been successful even between species [33,34,35]. Size regulation in these systems has been shown to be affected via changes to cell division, but we lack an understanding of the upstream sensation of population size in the early embryo. Importantly, chimeras occur naturally in human individuals, suggesting that the mechanisms of size regulation that we discuss may not solely relate to experimental systems but also to human health and fertility.

### 2.3. Homotypic Grafting Experiments

A homotypic graft may be described as the transplantation of a cell population from a donor embryo to the equivalent location in a host embryo—ultimately resulting in the production of a chimeric embryo if the host cells differ from donor cells in genotype. Homotypic grafts provide a powerful tool for fate mapping, provided that the donor tissue is labelled in some way such that it can be distinguished from host tissue. In its earliest form, this often involved utilising closely related species as host and donor that could be distinguished: for example, quail-chick chimeras were frequently used in early experimental embryology because quail cells could be identified through their cytology or using immunostaining [20]. Whilst quail nuclei contain conspicuous heterochromatic condensates associated with the nucleolus, chick nuclei instead have condensates throughout the nucleoplasm [36]. A similar approach has been utilised in amphibian experiments, with the difference in pigmentation between species lending itself to distinguishing host and donor tissue in grafts (see the next section on inductive reprogramming for more discussion on this point). More recently, transgenic lines have been constructed in many model organisms which allows for clear distinguishment of host and donor tissue under fluorescent imaging: this has the additional advantage of only involving cells of a single species, removing possible extraneous effects resulting from species differences, examples in [37,38]. This approach has the additional advantage of being compatible with live imaging approaches, unlike post-fixation staining (e.g., quail-chick chimeras). A particularly impressive use of the GFP transgenic chicken embryos for fate mapping is described by Solovieva et al. [2]. The Hensen’s node fate map [39] was revisited using single cell grafts to define the location of self-renewing stem cells, resulting in their precise localisation to the posterior portion of the node [2]. 

The remarkable regulative nature of the vertebrate embryo is ideal for fate mapping through adding cells to the embryo; normal development is rarely disrupted by such grafts, and they incorporate well into the host. In some cases, the host tissue is removed completely during this process but in others it is retained. Detailed fate maps for development have been produced in several species including chicken and *Xenopus* in part by homotypic grafting (as well as dye injections). In chicken for example, a huge research effort has employed experimental embryology to fate map the primitive streak stage embryo, revealing the organisation of different progenitor cells in and around the primitive streak [39,40,41,42,43,44,45,46,47]. In *Xenopus*, where injection of fluorescent dextran has been utilised extensively to fate map pre-gastrulation, transgenic lines have more recently been used such that analysis can be extended to post-metamorphosis stages (by which time fluorescent dextran has been diluted out) [37]. The contribution of such fate maps to our understanding of development should not be underestimated, because they provide fundamental insight into which regions of the embryo will contribute to which tissues and pave the way for more complex experiments investigating how these different populations are communicating with one another during development. 

### 2.4. Inductive Reprogramming

One of the most famous experiments in experimental embryology was performed by Spemann and Mangold (1924) [19] and defined the amphibian dorsal blastopore lip as the embryonic Organizer. Similarly, Saunders’ polarizing region grafts in the chick limb bud [48] are recognised for their dramatic impact on limb patterning, inducing a mirror image duplication of the digits in the AP axis. Through these pivotal experiments in embryology, we have subsequently come to learn—via a great deal of research effort—about the role of the signals deriving from these signalling centres in the development of the vertebrate primary body axis and limbs. 

The importance of these examples of adding cells to embryos should not be understated: though they are examples of highly localised signals that elicit particularly dramatic responses in host tissue upon grafting, they have pointed us toward fundamental principles of development—i.e., how do non-equivalent tissues interact to produce structures in the early embryo? Combining heterotypic grafts with tissue labelling (discussed in Section 2.3.) allows responses in host tissue to be distinguished from the autonomous production of embryonic structures from donor tissue. In the case of Spemann’s organizer grafts, use of differently pigmented *Triton* (newt) species was able to show that the ectopic organizer elicits a response by host tissue, leading to its incorporation into the secondary body axis [19]. This response involves ‘reprogramming’ of the normal, ventral fates that cells would take in a control embryo.

In following up the neural induction aspects of organiser function in *Xenopus* embryos, a critical study discovered region specific differences in the cell types produced from distinct regions of the archenteron roof, when transplanted using Einsteck grafts into the blastocoel of competent staged embryos. More medial regions were able to induce cell types corresponding to the central nervous system, lateral regions also generated cell types that are known to be derived from the neural crest, such as pigment cells [49]. The cloning of *snail2* (previously named *slug*), the first gene to mark early neural crest tissue, meant that such assays could be repeated and analysed at earlier stages post-grafting [50]. This enabled a much more in-depth study of the series of inductive interactions leading to neural crest specification during gastrulation and neurulation stages of development [51,52,53,54]. The advent of single cell sequencing and chromatin accessibility assays has taken this further, with a full description of the gene-regulatory network that responds to the known inductive interactions of early neural crest development [55].

While cell addition has been hugely informative in deciphering the mechanisms of cell fate specification during normal development, an additional set of embryological approaches has enabled researchers to probe the regulative capacity of pattern formation, which we will now turn to.

## 3. Cell Removal as an Essential Tool in Experimental Embryology

Removing cells from a developing tissue is a key part of an experimental embryologist’s tool kit. Cell removal is used here as an umbrella term for the removal of cells at any scale of organisation from single cells or groups of cells to whole populations or whole tissues. In addition, our definition encompasses both the complete excision of cells from their surroundings and their ablation within a tissue. It is important to note that where cells are ablated within a tissue, the presence of dead cells can potentially affect the response of the system differently from excision of the cells. Methods for cell removal range from surgical manipulation to laser-mediated ablation. In all permutations, cell removal can give insight into the necessity of the removed entity for a given phenotype. Cell removal can be further used to investigate the mechanisms of regulation in cases where the system is robust to cell loss. This makes it an invaluable starting point in breaking down the parameters of cell and tissue behaviour in a given system. Here, we highlight a range of studies that employ cell (or tissue) removal to great effect to begin to understand some of the major unanswered questions in developmental biology. We cover how this can be investigated at a range of length scales in the embryo from the cellular level to the tissue, and multi-tissue levels. 

Many classical pieces of experimental embryology used cell removal in early embryos to show that in some species regulation occurred and normal structures formed, while in others it did not. This contributed to the concepts of mosaic and regulative development. While useful to describe the result of manipulating a specific developmental process the majority of systems are neither wholly regulative nor wholly mosaic. This is illustrated in a recent update of the classic tradition of cell removal. The early cells of annelid embryos often have stereotypical fates and therefore are often associated with mosaicism. For example, in *Capitella telata*, two of the four primary micromeres generate the larval eyes. Micromere ablation results in the loss of the respective eye in roughly three quarters of embryos. However, the remaining embryos form two eyes successfully. The removal of both an eye and non-eye forming micromere increases the proportion of embryos missing an eye [4]. This indicates that the differentiation of larval eye cells has some regulative capacity and the ability to form eyes has not yet been completely specified during early development. 

Cell ablation has also been used to great effect to investigate cell behaviours in tissues at late development and adult stages. Many tissues have some regenerative capacity either in response to normal cell turnover or to injury. Genetically targeted ablation is often used to remove tissue resident stem cells. For example, the basal stem cells in the mouse airway epithelium, which express CK5, can be removed using a Tet-On diphtheria toxin transgenic line. On ablation of basal stem cells, secretory cells dedifferentiate and contribute to the regeneration of the stem cell population [5]. Notably, the amount of dedifferentiation is inversely proportional to the differentiation state of the secretory cells [5]. Cell removal studies can shine light onto changes in cell behaviour other than differentiation state. The *Xenopus* tadpole tail can regenerate at certain developmental stages following amputation. However, this regeneration is not accompanied by the emergence of a population with a multipotent signature or evidence of transdifferentiation between cell types. Instead, it appears that a unique population of cells directs the proliferation of tissue specific progenitors in order to orchestrate accurate regeneration [6]. Furthermore, when these regeneration-organizing cells are genetically ablated using the nitro-reductase system, tail regeneration is reduced [6]. The key cell behaviours that drive the regulation of this system are therefore the migration and signalling of the regeneration-organizing cells and the proliferation of the other progenitors.

A long standing and complex question in developmental biology is how patterns scale with changes in the size of the tissue, the details of which have been recently reviewed [56]. This is important from an evolutionary perspective where there is variation in tissue size, but conservation in patterning mechanism, between species. An excellent example of the robustness of patterning is the removal of cells from the early zebrafish embryo. Following removal of 30% of the zebrafish blastula, a normally patterned embryo is formed after gastrulation [7]. This is because the specification of the mesoderm scales in response to size-reduction creating germ-layers with correct proportions. By combining mathematical modelling and in vivo protein anchoring the authors predicted and demonstrated that the scaling of the pattern is due to the high diffusivity of the Nodal-repressor Lefty. Lefty is expressed at the blastoderm margin under high Nodal levels but diffuses faster than Nodal. Further from the margin Lefty concentration is high enough to inhibit Nodal and therefore limit mesendoderm specification. In an embryo with a reduced blastoderm a high concentration and diffusivity of Lefty is essential to that Lefty builds up faster and restricts the mesoderm to fewer cells [7]. Other studies have utilised a blastoderm reduction technique in combination with mathematical modelling and quantitative imaging to investigate somite scaling [8] and dorso-ventral Bone Morphogenetic Protein (BMP) gradient scaling [9].

Classical embryological experiments involving the bisection of the chick epiblast have long provided insight into the highly regulative nature of primitive streak formation in the early chick embryo [57,58]. A recent investigation in quails has combined laser-mediated bisection of the embryo with high resolution imaging to demonstrate that mechanical signals scale robustly following tissue removal [10] (Caldarelli et al., 2021; biorxiv). The formation of the primitive streak in the chick embryo is well documented to involve large scale tissue flows (reviewed in [59]). Quantification of these movements in both chick and quail indicates that tissue contractility in the posterior of the embryo drives the formation of the primitive streak. This contraction must be balanced by tension from the embryonic margin in order to restrict the localisation of the streak [10,60]. Following bisection, the amount of embryonic margin has decreased, which increases the tension across the tissue and decreases the size of the contractile domain. This facilitates scaling of the primitive streak and associated gene expression in the half-embryo [10]. These studies demonstrate the importance of combining cell removal techniques with modern quantification in order to understand the parameters of tissue patterning and signaling feedback.

At a multi-tissue level, the development of an individual tissue can influence the correct development of surrounding tissues. Separating tissues can be used to pick apart these interactions. It has been shown that the looping morphogenesis of the gut tube is dependent on the attached dorsal mesentery tissue. Complete separation of the two tissues ex ovo, as well as partial removal of the mesentery in ovo, result in the loss of gut loops [11]. The relaxation of the gut indicates that it was under compression, while the mesentery tissue contracts suggesting that it was under tension. This experiment allowed the authors to distinguish between potential hypotheses of what causes gut looping and ultimately demonstrate that it is driven by differential tissue growth [11]. Removing whole tissues can also be used to understand how tissues signal to one another. Plants provide an excellent model system for this as they are highly adapted to dealing with loss of tissue. In one study, the laser ablation of a tomato leaf primordia resulted in a new primordia forming near the removal site, indicating that a forming leaf primordia has an inhibitory effect on the surrounding tissue in order to ensure minimal shading between leaves [12]. Removing previously established primordia further highlighted how the exact position of a future primordium relies on signals from several different established primordia [12].

The removal of groups of cells can also have multi-tissue effects. The maturation of the notochord is essential for correct elongation of other axial tissues in the formation of the zebrafish tail. In normal development anterior notochord cells expand through vacuolation, while new cells are added from a progenitor pool in the tailbud. Ablation of notochord cells in both the anterior and posterior, followed by quantification of tissue changes from live imaging data, demonstrates how the notochord effects the elongation of the adjacent somites without affecting the formation of new somites [13,14]. Overall, this affects the elongation of the tail [13]). Similarly, to the gut and mesentery above, the notochord and somites are mechanically coupled through the extra-cellular matrix [61]. This highlights how removing cells from one tissue can affect the development of adjacent tissues via multi-tissue mechanical coupling.

## 4. Tissue Embedding as an Essential Tool in Experimental Embryology

Part of the experimental embryology toolbox is dedicated to performing mechanical manipulations of the embryos without changing the number of cells, such as compressing embryos between two coverslips [62]. An emerging approach to control the mechanical inputs that affect embryos, explants or organoids is embedding them in 3D substrates of known properties. These new approaches allow controlling the physical and chemical characteristics of the substrate separately, as well as quantifying the force exerted by developing tissues. Embedding can help to study complex phenotypes affecting force generation, distinguish intrinsic and extrinsic morphogenetical cues, and dissect the interactions of mechanical and biochemical signals. To illuminate advances in this emerging quantitative experimental embryology approach we will briefly review some key examples from recent research into understanding the role of mechanical forces in guiding developmental processes.

### 4.1. Force Generation and Tissue Mechanics during Development

Morphogenesis in developing tissues is simultaneously affected by the amount of force generated and the passive mechanical parameters, such as stiffness. For this reason, it is challenging to differentiate when specific cellular or molecular processes affects force production or changes the tissue’s passive mechanical properties. 

Rho kinase (Rok) is an essential regulator of actomyosin contractility with an important role in different morphogenetic processes [63]. Surprisingly, when [64] chemically inhibited Rok in developing frog embryos, it did not produce a visible phenotype, although it significantly reduced dorsal tissue stiffness. They proposed three plausible explanations: (1) Rok is unimportant for the process of convergence and extension (CE), (2) Other molecular and cellular processes compensate for the effects of Rok inhibition in force generation or (3) Rok inhibition reduces force generation, compensated by a concomitant reduction in tissue stiffness. To distinguish among these three possibilities [15] developed an embedding-based method to assess force generation during the CE of *Xenopus* explants. They quantified the bulk force generated by explants embedded in agarose gels of known viscoelastic properties by measuring the displacement of fluorescent micro-beads dissolved in the substrate using quantitative image analysis. Unconstraint dorsal explants treated with a RoK inhibitor do not show significant differences in CE; however, they fail to elongate and generate significantly less force under the mechanical load imposed by the agarose gel when compared with controls. This shows that RoK inhibition reduces tissue stiffness and force production, producing a situation where the tissue can elongate if unchallenged. Finally, using the same technique, they show that different explants where the notochord was surgically removed or mock-operated have the same force-generating capacities, suggesting that the notochord is unnecessary for tissue elongation at this point in development, in contrast to what has been observed in zebrafish embryos at later stages (see above).

More recently, a similar study [16] used agarose embedding to demonstrate that the Armadillo Repeat gene was deleted in Velo-Cardio-Facial syndrome (ARVCF), a member of the catenin family, which is necessary for force generation. Arvcf Knock Down (KD) embryos have defective anteroposterior elongation with shorter and bent bodies, which suggests a deficient CE. To test the presence of CE defects, they explanted dorsal tissues from control and Arvcf KD embryos. Surprisingly, Arvcf KD dorsal explants did not manifest quantifiable differences in elongation compared to controls. They hypothesised that Arvcf could be important during CE as the level of force necessary to overcome the resistance from the surrounding tissues in the embryo. To test this hypothesis, they recreated the mechanical load imposed by the other tissues by embedding the explants in soft agarose. In these conditions, control explants extend effectively while the Arvcf KD dorsal explants fail to elongate, buckle as seen in the embryo, and generate significantly less force than controls. Finally, they used the buckling of the embedded tissue to estimate its stiffness. Calculating the buckling showed that the tissue compliance increased significantly in Arvcf KD dorsal explants.

Together, these studies show how embedding allows studying the effects of complex phenotypes that simultaneously affect force generation and the mechanical properties of the tissues.

### 4.2. Intrinsic and Extrinsic Mechanical Cues in Development

Developing tissues can generate intrinsic mechanical cues to guide morphogenesis or differentiation. These intrinsic signals can be substituted by extrinsic mechanical signals, as shown by the rescue of tissue invagination in myosin mutant drosophila embryos [65]. However, these external mechanical triggers can also be utilised by developing organisms. This is probably especially relevant in plants, where development does not occur in an egg isolated from environmental, mechanical inputs.

In dicotyledonous plants, differential cell elongation in the hypocotyl generates a hook that protects the meristems from the soil during germination. Baral et al. [66] noted that *Arabidopsis* loss-of-function katanin mutants -a protein that prevents branching and promotes bundling of microtubules- cannot generate the apical hook when the seedlings are germinated on the surface of the agar. However, when they sprouted the seedling inside sterile soil, a more natural condition, they rescued the formation of the hypocotyl hook. To distinguish between the biochemical and mechanical constraints imposed on the soil, they germinated the seedlings inside a soft-agarose substrate. This condition also rescued the formation of the hypocotyl hook, showing that it can be triggered exclusively by external mechanical signals. Later, using a combination of pharmacological and genetic perturbations [66] showed how the mechanical cues from the soil can trigger the reorganisation of microtubules and, ultimately, tissue folding independently of katanin. This way, they revealed that the hypocotyl hook could be triggered by a katanin-dependent intrinsic pathway and an extrinsic mechanically dependent pathway. Both pathways probably contribute to the robust formation of the hook.

A similar use of agarose gels can be used to study the requirement of external forces in animal development. Hiramatsu et al. [67] cultured early mouse embryos in agarose gels of increasing stiffness. They found that the proper elongation of the conceptus needs a stiff enough external environment. A series of complementary experiments showed that the external mechanical environment triggers the differentiation of the Distal Visceral Endoderm (DVE), suggesting the physical properties of the uterine tissues. The degree to which alterations in the mechanical constraint imposed by extra-embryonic tissues impact the development and evolution of gastrulation morphogenesis has been recently reviewed elsewhere [68]. Advances in ex utero culture and imaging of mammalian embryos will likely lead to exciting new directions for this research agenda [69]. 

### 4.3. Control of Mechanical and Biochemical Parameters

Extrinsic mechanical and biochemical signals from adjacent tissues or the extra-cellular matric (ECM), critically affect morphogenesis and differentiation. Unfortunately, it is challenging to discriminate the effect of external specific mechanical and biochemical parameters on different biological processes. However, new generations of hydrogels allow generating chemically defined 3D substrates where the physical and biochemical properties can be modified independently of each other.

A paradigmatic example of such an approach can be found in [17]. Ranga et al. tested how external conditions affected the generation of neural tube organoids using a PEG-based hydrogel where the stiffness, degradability and biochemical composition can be orthogonally modified in combinatoric way. Using automatic image analysis, they studied how different biochemical and mechanical variations affected other aspects of development, such as patterning, proliferation, or the acquisition of cell polarity. For instance, they show that the dorsoventral patterning is enhanced independently by an intermediate stiffness, a non-degradable matrix and by, to a minor degree, laminin. Furthermore, using inhibitors and activators of the cytoskeleton contractility, they found evidence supporting that, in this case, intrinsic mechanical signals produced by the cells cannot be overwritten by extrinsic forces generated by the stiffer substrates. 

In a more recent example, Elosegui-Artola et al. [18] developed alginate hydrogels conjugated with integrin-binding sites (Arg-Gly-Asp peptide) where the viscoelastic properties can be changed independently of the stiffness, pore size and adhesive ligands. Using these gels in combination with computer simulations, they show how the formation of fingering morphologies of human breast epithelia and intestinal organoids depends on the substrate’s and tissue’s relative mechanical parameters. Similarly, hydrogels have been developed that enable the independent alteration of both gel stiffness and ECM composition to explore their relative impact on cells cultured on specific substrates [70]. 

The role of mechanical cues in guiding cell migration has been recently studied in the context of cell migration during embryonic development [71]. By taking primary explants of the cranial neural crest from the *Xenopus* embryo and culturing them on substrates of different stiffness, it was shown increasing the stiffness of the substrate increased efficiency of chemotaxis [72]. Importantly, this increasing substrate stiffness was evident in vivo, as the underlying mesodermal tissue increases its apparent elasticity at the onset of neural crest migration [72]. Overlying the neural crest is a pan-placodal domain, that splits into distinct cranial placodes at the onset of neural crest migration [73], and undergoes a chase-and-run interaction to help guide neural crest migration [74]. This interaction has also recently been shown to involve mechanical forces, as the neural crest imposes a stiffness gradient in the responding placodal population, synergizing with chemotaxis at the molecular level [75]. Understanding how multiple mechanisms integrate to drive the directional migration of cells is an exciting new field, that incuses a consideration of how electrical signals may interact with mechanical cues [76]. The ease at which cell and tissue explants can be generated through experimental embryology is at the centre of this approach. 

## 5. Conclusions

A central feature of embryonic development is its ability to regulate upon cell or tissue loss and to go on to generate well patterned organs and organisms. The self-organisation feature of development has long been known, but still not well understood. Here, we point to experimental embryology as the central method by which to probe this process, in combination with advanced quantitative methods of both molecular biology and imaging. We suggest that Quantitative Experimental Embryology should be situated as modern classical approach that is at the very forefront of developmental and stem cell biology.

## Figures and Tables

**Figure 1 jdb-10-00044-f001:**
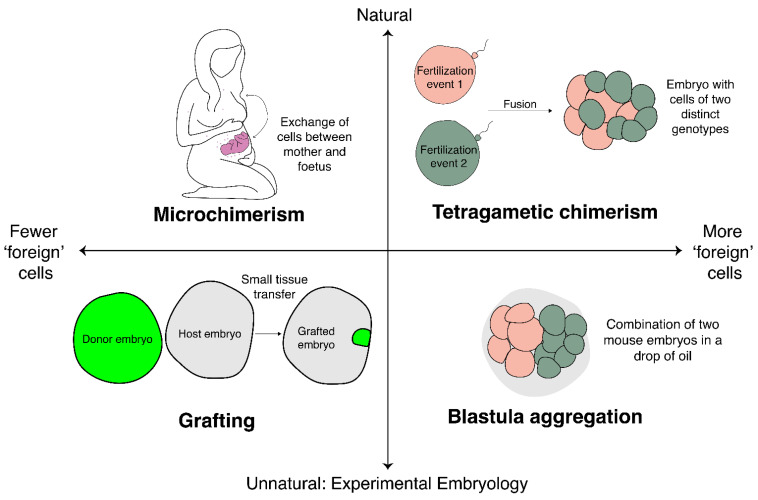
A spectrum of chimeras. This diagram conceptualises examples of chimeras discussed across two axes: degree of naturality (i.e., naturally occurring or produced by experimental manipulation), and the proportion of foreign cells relative to total cell number. Naturally occurring chimeras include microchimeras (**top left**) and tetragametic chimeras (**top right**). Experimentally produced chimeras include grafted embryos (**bottom left**) and blastula aggregation experiments (**bottom right**). See text for details.

**Table 1 jdb-10-00044-t001:** Experimental embryology techniques and examples of relevant biological questions that they can help to elucidate.

Experimental Embryology Techniques	Questions
Adding cells	Scaling
*Homotypic grafts*	Cell competition
*Heterotypic grafts*	Cell autonomous vs. non cell autonomous processes
*Embryonic aggregates*	Fate mapping
	Inductive reprogramming
Removing cells	Regeneration
*Single cell removal*	Scaling
*Genetically targeted ablation*	Mechanical regulation
*Tissue dissection*	Multi tissue coupling
	Competence
Confining cells	Force generation
*Agarose gels*	Intrinsic vs. extrinsic mechanical signals
*Matrigel*	Mechanical versus biochemical signals
*Biochemically & Mechanically defined Hydrogels*	Force adaptation

**Table 2 jdb-10-00044-t002:** Examples of modern quantitative adaptations of experimental embryology techniques.

Paper	Experimental Embryology Method	Modern/Quantitative Addition
Solovieva et al., 2022 [2]	Grafting	Single cell RNA-sequencing and live imaging
Guibentif et al., 2021 [3]	Chimera production (blastula aggregation)	Single cell RNA-sequencing
Yamaguchi et al., 2016 [4]	Single cell ablations	Used laser ablation.
Tata et al., 2013 [5]; Aztekin et al., 2019 [6]	Cell population ablation	Genetically-targeted ablation. (Tet-On diptheria toxin; Nitroreductase).
Almuedo-Castillo, et al., 2018 [7]; Ishimatsu et al., 2018 [8]; Huang and Umulis, 2019 [9]	Tissue removal	Used classical methods to remove cells (capillary; hairloop; needle). Combined with imaging and mathematical modelling to predict how the system scales and other perturbations, such as protein-anchoring, to test the model’s predictions.
Caldarelli et al., 2021 [10]	Tissue removal (embryo bisection)	Used laser ablation. Coupled with high-resolution imaging to quantify the mechanical forces of tissue movement.
Savin et al., 2011 [11]	Tissue removal	Used classical tissue dissection followed by mathematical modelling of tissue properties.
Reinhardt et al., 2005 [12]	Tissue ablation	Used laser ablation.
McLaren and Steventon 2021 [13]; Ozelci et al., 2022 [14]	Tissue ablation	Used laser ablation. Coupled with high resolution live imaging.
Zhou et al., 2015 [15]; Huebner et al., 2022 [16]	Embedding	Quantification of force generation (displacement of fluorescent micro-beads and tissue buckling)
Ranga et al., 2016 [17]	Embedding	Robotics and automatic image analysis
Elosegui-Artola et al., 2022 [18]	Embedding	Computer simulations to study how fingering behaviour depends on the mechanics of the substrate

## Data Availability

Not applicable.

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
