# Peer review of "Quantitative Experimental Embryology: A Modern Classical Approach"

_jdb, 2022, doi:10.3390/jdb10040044_

Round 1

Reviewer 1 Report

Quantitative Experimental Embryology: A Modern Classical Approach

Busby et al. present a very timely discussion of the key contributions made by the application of long standing (“classical”) experimental embryology techniques towards answering some of the most fundamental questions in Developmental Biology. They convincingly make the case that these have been enhanced – rather than replaced – by the appearance of molecular biology/genetic quantitative approaches. The authors revisit key studies that exemplify the utility of such applications to respond to some of the current burning questions in the Developmental Biology field.

The manuscript is well written and will be very useful for the scientific community working in both Developmental and Stem Cell Biology. Minor changes are suggested to improve the final version for publication.

Minor issues:

·         Topic organization throughout the text is confusing. A more coherent use of text format (“bold” and/or topic numbering) should be employed so the reader can easily follow the author’s intended organization. For example, formatting of the subtopic 1A title (line 121) is equivalent to that of the main title in line 58, so it is not readily perceived as a subtitle of point 1 (line 76). Also, numbering is employed in the subtitles in lines 76 and 254, but not in equivalent subtitles in lines 413, 460 and 495.

·         The proposed clustering of the main techniques addressed into three major groups (cell/tissue addition, removal, and conditioning) is very useful and comprehensive. However, while the authors present the latter as a classical approach in the beginning of the manuscript (“…altering the mechanical environment of cells through in vitro cell culture and tissue confinement”, line 43), they summarize this concept in the subtitle of “Tissue embedding…” in line 403, where they present it as an “emerging” tool. Some clarification of the classical vs modern tools for cell/tissue conditioning should be provided.

·         A Table summarizing the classical approach and the corresponding modern/quantitative addition in each of the studies cited would greatly benefit the manuscript. In some of the experimental studies presented, it is not very clear what the authors are considering as the “classical” vs “modern” approach.

·         Some additional classical experiments of “Cell removal”, for example in the chick embryo, could be worth mentioning because of the critical biological insights provided. See for example Bellairs et al. 1967 (https://doi.org/10.1242/dev.17.1.195), Spratt 1947 (https://doi.org/10.1002/jez.1401040105), Spratt 1942 (https://doi.org/10.1002/jez.1400890104), Waddington 1932 (https://doi.org/10.1098/rstb.1932.0003), to name a few.

·         In addition to the classical approaches for cell fate mapping in the chicken embryo (referenced in lines 233-235), the recent work using GFP transgenic chickens to revisit the Hensen’s node fate map may be worth mentioning, https://doi.org/10.1016/j.ydbio.2022.06.015.

·         The Conclusions section mentions the “advanced quantitative methods of both molecular biology and imaging”. The contribution of modern imaging techniques to the analysis and interpretation of classical experimental approaches, however, was not highlighted throughout the text and would merit some attention.

·         Dunsford et al., 1953 mentioned in line 85 is not listed in the References.

·         A Reference seems to be missing at the end of line 88 to support the previous statements.

·         “…reduction to technique…” should read “…reduction technique…” (line 353)

·         A word seems to be missing in line 461: “Developing tissues can generate intrinsic mechanical ____ to guide morphogenesis”.

·         Line 481: “…katanin-depend…” should read “…katanin-dependent…”

·         Line 484: “…culture early mouse…” should read “… cultured early mouse…”

·         Line 498: “…specific mechanical parameters and biochemical…” should read “…specific mechanical and biochemical parameters…”

·         Line 526: “By talking…” should read “By taking…”

·         Line 528: “…stiffness of thee substrate…” should read “…stiffness of the substrate…”

Author Response

Minor issues:

  • Topic organization throughout the text is confusing. A more coherent use of text format (“bold” and/or topic numbering) should be employed so the reader can easily follow the author’s intended organization. For example, formatting of the subtopic 1A title (line 121) is equivalent to that of the main title in line 58, so it is not readily perceived as a subtitle of point 1 (line 76). Also, numbering is employed in the subtitles in lines 76 and 254, but not in equivalent subtitles in lines 413, 460 and 495.

We thank the reviewer for bringing this to our attention - we have reformatted all headings in the manuscript to be more consistent.

  • The proposed clustering of the main techniques addressed into three major groups (cell/tissue addition, removal, and conditioning) is very useful and comprehensive. However, while the authors present the latter as a classical approach in the beginning of the manuscript (“…altering the mechanical environment of cells through in vitro cell culture and tissue confinement”, line 43), they summarize this concept in the subtitle of “Tissue embedding…” in line 403, where they present it as an “emerging” tool. Some clarification of the classical vs modern tools for cell/tissue conditioning should be provided.

We added more context in the introduction to this section. Mechanical perturbations that do not change the number of cells have been part of classic experimental embryology (Driesch, 1892). We clarify that the use of gel embedding is a modern approach to modify the external mechanical environment. Lines 417-428.

  •      A Table summarizing the classical approach and the corresponding modern/quantitative addition in each of the studies cited would greatly benefit the manuscript. In some of the experimental studies presented, it is not very clear what the authors are considering as the “classical” vs “modern” approach.

We have added a table to the end of the manuscript which addresses this point.

  • Some additional classical experiments of “Cell removal”, for example in the chick embryo, could be worth mentioning because of the critical biological insights provided. See for example Bellairs et al. 1967 (https://doi.org/10.1242/dev.17.1.195), Spratt 1947 (https://doi.org/10.1002/jez.1401040105), Spratt 1942 (https://doi.org/10.1002/jez.1400890104), Waddington 1932 (https://doi.org/10.1098/rstb.1932.0003), to name a few.

We agree with the reviewer and have cited the work of Spratt 1942, and Spratt and Haas, 1960 on line 358 as these studies directly pertain to the following paragraph on recent quail bisection experiments.

  • In addition to the classical approaches for cell fate mapping in the chicken embryo (referenced in lines 233-235), the recent work using GFP transgenic chickens to revisit the Hensen’s node fate map may be worth mentioning, https://doi.org/10.1016/j.ydbio.2022.06.015.

Thank you - we have added a few sentences describing this work.

  • The Conclusions section mentions the “advanced quantitative methods of both molecular biology and imaging”. The contribution of modern imaging techniques to the analysis and interpretation of classical experimental approaches, however, was not highlighted throughout the text and would merit some attention.

We thank the reviewer for this comment and have endeavoured to include signposting of where modern imaging techniques were used throughout the text.

  • Dunsford et al., 1953 mentioned in line 85 is not listed in the References.

We have added this reference, thank you for bringing this to our attention.

  • A Reference seems to be missing at the end of line 88 to support the previous statements.

We have added a reference to support the previous statements.

  • “…reduction to technique…” should read “…reduction technique…” (line 353)

Corrected

  • A word seems to be missing in line 461: “Developing tissues can generate intrinsic mechanical ____ to guide morphogenesis”.

Corrected, added ‘cues’.

  • Line 481: “…katanin-depend…” should read “…katanin-dependent…”

Corrected

  • Line 484: “…culture early mouse…” should read “… cultured early mouse…”

Corrected

  • Line 498: “…specific mechanical parameters and biochemical…” should read “…specific mechanical and biochemical parameters…”

Corrected

  • Line 526: “By talking…” should read “By taking…”

Corrected

  • Line 528: “…stiffness of thee substrate…” should read “…stiffness of the substrate…”

Corrected.

Reviewer 2 Report

This review is well-structured and detailed, describing the history and development of Experimental Embryology.

For the convenience of readers, please make the following improvements

1)      On Line 145:  please give full name of “ES cells”

2)      On Line 354:  please give full name of “BMP”

3)      On Line 496:  please give full name of “ECM”

4)      On Line 519:  please mention the breast epithelial cells from which species.

5)      On line 504: “They tested how 1000 external conditions..” please read the cited article again.

6)      On line 349: It would be great, if you describe a little more about “Lefty”.

Author Response

1)      On Line 145:  please give full name of “ES cells”

Corrected.

2)      On Line 354:  please give full name of “BMP”

Corrected.

3)      On Line 496:  please give full name of “ECM”

Corrected.

4)      On Line 519:  please mention the breast epithelial cells from which species.

Corrected. Added ‘human’.

5)      On line 504: “They tested how 1000 external conditions..” please read the cited article again.

Corrected. The exact number conditions are not reported in the paper. We clarify that they vary the experimental conditions in a combinatory way.

6)      On line 349: It would be great, if you describe a little more about “Lefty”.

We agree with the reviewer and have included additional sentences describing how the nodal-lefty inhibition-activation system is thought to operate.

Reviewer 3 Report

This review by Lara Busby and colleagues within the Steventon lab is very nicely written. It has clear sections and sub headings. It nicely ties in recent studies with classic experiments and indirectly pays homage to Roberto Mayor's work and impact. It would be good to include a small section on the NCCs as they are such key cells in what Roberto works on and so much of the collective migration work done by Shellard et al and Barriga et al were done on the xenopus NCC. Although so much positive is mentioned about chimeras, it would be worth noting or explaining when chimeras can also go wrong and how this affects development. If possible, it would be useful to readers to see schematics of the different sections - such as the grafting, ablations etc. 

Author Response

It would be good to include a small section on the NCCs as they are such key cells in what Roberto works on and so much of the collective migration work done by Shellard et al and Barriga et al were done on the xenopus NCC.

These papers have been cited in lines 542-559.

Although so much positive is mentioned about chimeras, it would be worth noting or explaining when chimeras can also go wrong and how this affects development.

We have added a sentence describing possible negative effects of human chimerism in line 89.

If possible, it would be useful to readers to see schematics of the different sections - such as the grafting, ablations etc.

We added two tables that summarise what type of questions experimental embryology can answer and how they can be combined with modern approaches. We appreciate this comment but do not believe that a schematic would add substantially to the clarity of the review.